# Point-of-care creatinine-based eGFR (StatSensor) in detecting kidney dysfunction (KD) among people living with HIV in Tanzania

Daniel Msilanga[1,2]*, Anthony Muiru[3], Elizabeth Msangi[1,2], Jacqueline Shoo[1,2], Jonathan Mngumi[1,2], Ewaldo Komba[1,2], Emmanuel Balandya[1], Paschal Ruggajo[4], Rajendra Bhimma[5], Kathleen Liu[3]

**1** Muhimbili University of Health and Allied Sciences, Dar es Salaam, Tanzania, **2** Muhimbili National Hospital, Tanzania, **3** University of California San Francisco, California, United States of America, **4** The Aga Khan University, Dar es Salaam, Tanzania, **5** University of KwaZulu-Natal, South Africa

* pascodanny@gmail.com

## Abstract

### Introduction

Kidney Dysfunction (KD) is prevalent among people living with HIV (PLHIV) in low- and middle-income countries (LMICs), but routine screening is limited due to inadequate laboratory infrastructure. The StatSensor® Point-of-Care (POC) Creatinine Test offers a rapid, cost-effective alternative for early KD detection, though its accuracy in PLHIV remains uncertain.

### Methods

We conducted a diagnostic accuracy cross-sectional study at Temeke Regional Referral Hospital (TRRH) HIV Clinic from January to March 2025 among PLHIV aged ≥18 years. Kidney dysfunction (KD) was defined as an estimated glomerular filtration rate (eGFR) <60 mL/min/1.73 m² using the CKD-EPI 2021 equation. We compared StatSensor point-of-care eGFR results with eGFR derived from serum creatinine measured by the Jaffe method. Diagnostic performance metrics including sensitivity, specificity, predictive values, and receiver operating characteristic (ROC) curves were reported.

### Results

Among 358 participants, the median age was 48 years, with 66.2% female and 15.6% having KD (eGFR<60 mL/min/1.73m²). The StatSensor demonstrated 92.9% sensitivity, 94.7% specificity, and 94.4% overall diagnostic accuracy compared to the Jaffe method. The ROC curve (AUC=0.938) indicated strong test performance, showing substantial agreement with a kappa value of 0.805. Bland-Altman analysis

**Data availability statement:** All relevant data are within the paper and its Supporting Information files.

**Funding:** This project was funded by the Fogarty International Center of the National Institutes of Health (NIH) Award Number D43TW009343 and the University of California Global Health Institute (UCGHI). The funders had no role in study design, data collection and analysis, decision to publish, or preparation of the manuscript.

**Competing interests:** The authors have declared that no competing interests exist.

revealed a negative bias of 4.36 mL/min/1.73 m² with limits of agreement from −19.68 to 28.40 and a strong correlation (R² = 0.813) between the two methods.

## Conclusion

The StatSensor POC Creatinine test demonstrated high diagnostic accuracy and strong agreement with the standard Jaffe method, indicating its potential as a reliable screening tool for kidney dysfunction in PLHIV in resource-limited settings.

## Introduction

Kidney dysfunction (KD) is a significant concern among people living with HIV (PLHIV) in low-middle-income countries (LMICs), with prevalence rates ranging from 10% to 35% [1–3]. Despite guideline recommendations for routine screening, implementation remains limited, particularly in primary healthcare settings, where nearly 70% of people seek care. It is estimated that only one in three patients receives recommended creatinine testing (2). This is largely due to inadequate laboratory infrastructure [4–6]As a result, many patients are not diagnosed in the early stages of KD, and patients are typically identified only when they present with symptoms of end stage kidney failure (ESKD), often requiring hemodialysis [7–9]. There is an urgent need for a feasible, and accessible test, to improve early detection and management of KD in PLHIV [9].

The StatSensor® POC Creatinine test (by Nova Biomedical Waltham, MA, USA), is a handheld point-of-care test designed to overcome the limitations of traditional laboratory creatinine testing [10,11]. This user-friendly and device requires minimal training and provides rapid results using capillary/venous blood from a finger stick, enabling early identification of KD [11]. Several studies on the StatSensor Creatinine test have shown variable findings, including sensitivity and specificity above 90% and a positive mean eGFR bias compared to laboratory-based creatinine measurements by Jaffe method [10,11–14]. While this test presents a promising option for KD screening, its performance variability across different populations warrants further evaluation, and its utility as a screening tool for KD in PLHIV in resource-limited settings remains unexplored [9].

To address the gap in KD screening among PLHIV in Tanzania and similar resource-limited settings, this study aimed to evaluate the diagnostic accuracy of the StatSensor Creatinine test in detecting KD among PLHIV receiving HIV care. By comparing its performance with, conventional laboratory-based serum creatinine test, this study aimed to provide critical insights into its potential as a reliable and cost-effective screening tool in settings with limited laboratory capacity.

## Methods

### Ethical approval

The study was reviewed and approved by the National Institute for Medical Research (NIMR) under reference number NIMR/HQ/R.8a/Vol.IX/4695 and permission to

conduct the study was given by Temeke Regional Referral Hospital. The study objectives, methods, procedures, and potential risks were thoroughly explained to participants before obtaining their consent. All participants provided written informed consent before enrollment and granted permission before undergoing any study procedures.

## Study design and settings

We conducted a diagnostic accuracy cross-sectional study at Temeke Regional Referral Hospital (TRRH) HIV Clinic from 5th January to 30th March 2025. TRRH is one of the largest HIV care centers in the region, with 24,000–30,000 study visits per year, and a catchment area of 1,346,674 resident of Tanzania.

## Recruitment procedure

We included people living with HIV (PLHIV) aged 18 years or older who were receiving care at TRRH during the study period. Participants were selected daily using Probability Proportional to Size (PPS) sampling based on the clinic's appointment register to ensure representation across varying clinic volumes. Each eligible patient was assigned a unique study ID, and recruitment continued daily until the target sample size was reached.

Kidney dysfunction was defined as eGFR $<60$ mL/min/1.73m², calculated using the CKD-EPI 2021 equation (2,17). All eligible patients were informed about the study, and those who provided written informed consent were enrolled. Data were collected using a structured, interviewer-administered electronic questionnaire, which included socio-demographic factors, clinical history, and selected laboratory results.

Blood samples were collected by trained phlebotomists using standard aseptic techniques via venipuncture. A total of 5 mL of whole blood was drawn from each participant. A drop of whole blood was immediately analyzed at the bedside using the Nova StatSensor Xpress Creatinine Point-of-Care Meter (Nova Biomedical, Waltham, MA, USA). The device employs multi-enzyme amperometric biosensor technology to measure creatinine levels in whole blood, with a coefficient of variation (CV) ranging from 4.4% to 5.8% at clinical decision levels [10]. Calibration was factory-encoded in the reagent strip, and trained technicians performed all measurements according to the manufacturer's instructions. The technicians were blinded to the results of the reference method.

The remaining blood sample was placed in a non-heparinized serum-separating tube (SST), securely sealed to minimize air exposure, and stored in a cool box. Samples were transported to the central laboratory within one hour of collection and centrifuged at 3000G. Serum creatinine was measured using the Jaffe method on the Cobas Integra 400 Plus analyzer (Roche Diagnostics), which is traceable to isotope dilution mass spectrometry (IDMS). The assay demonstrated coefficients of variation (CVs) of 2.2% at 88 µmol/L and 1.8% at 482 µmol/L, based on internal quality control data. The laboratory participates in an External Quality Assurance (EQA) program coordinated by, and accredited to, the National Accreditation Board for Testing and Calibration Laboratories (NABL) and the College of American Pathologists (CAP).

## Statistical analysis

All analyses were conducted using STATA 17. Descriptive statistics included measures of central tendency (mean, median) and dispersion (standard deviation, quartiles, and interquartile range). StatSensor creatinine values were compared with laboratory-based Jaffe creatinine measurements using ordinary least squares linear regression and Bland–Altman analysis. The regression analysis included estimation of the slope, intercept, and coefficient of determination ($R^2$), with the laboratory Jaffe method serving as the reference. In the Bland–Altman analysis, the mean bias and 95% limits of agreement were calculated, and comparisons were assessed against the allowable total error (TEa) margin of ±15%, as recommended by Clinical Laboratory Standards Institute (CLSI). The diagnostic performance of the StatSensor was further evaluated through sensitivity, specificity, positive predictive value (PPV), and negative predictive value (NPV), calculated using two-by-two contingency tables with the reference method as the gold standard. Receiver Operating

Characteristic (ROC) curve analysis was performed to determine overall diagnostic accuracy, with the Area Under the Curve (AUC) reported. Likelihood ratios and diagnostic odds ratios were also calculated. Pearson's correlation coefficient was used to assess linear relationships between continuous creatinine values, and simple linear regression was used to evaluate the association between StatSensor-derived eGFR and laboratory-measured creatinine. A p-value <0.05 was considered statistically significant.

## Results

A total of 358 participants were included in the study. The median age was 48 years (IQR: 39–54), with 56.9% of participants under the age of 50, and 66.2% were female. The median duration of HIV was 132 months (IQR: 84–180), with most participants on a tenofovir disoproxil fumarate (TDF)-based regimen (91.1%). A total of 15.6% had reduced eGFR (<60 mL/min/1.73m²), and the median estimated eGFR using the Jaffe method was 87.0 mL/min/1.73m² (IQR: 68.8–104.0), and **80.0 mL/min/1.73m² (IQR: 65.0, 101.4) by Statsensor method (Table 1).**

Among the 358 participants, the Stat-Sensor correctly identified 52 out of 56 participants with low eGFR, while 4 were misclassified as normal. Additionally, 286 out of 302 participants were correctly classified as having normal eGFR, whereas 16 were misclassified as having low eGFR (Table 2)

**Table 1. Socio-demographic and clinical characteristics of the study participants, n = 358.**

| Variable | Frequency (n) | Percent (%) |
|---|---|---|
| Age group (years) | | |
| < 50 | 204 | 56.9 |
| ≥ 50 | 154 | 43.1 |
| Median age in years (IQR) | 48 (39, 54) | |
| Gender | | |
| Male | 121 | 33.8 |
| Female | 237 | 66.2 |
| Median duration of HIV in months (IQR) | 132 (84, 180) | |
| Type of ART | | |
| TDF Based Regimen | 326 | 91.1 |
| Non TDF Based Regimen | 32 | 8.9 |
| Median creatinine (Jaffe method) (IQR) µmol/L | 80.0 68.0 to 94.75 | |
| Median creatinine (Statsensor method) (IQR) µmol/L | 85.5 (71.0 to 101.0) | |
| Estimated eGFR (Jaffe Method) | | |
| < 60 | 56 | 15.6 |
| ≥ 60 | 302 | 84.4 |
| Median standard eGFR (Jaffe method) (IQR) | 87.0 (68.8, 104.0) | |
| Median and IQR eGFR (Statsensor) | 80.0 (65.0, 101.4) | |

TDF-Tenofovir Disoproxil Fumarate, eGFR-Estimated glomerular filtration rate, ART-Antiretroviral therapy

**Table 2. Diagnostic Comparison of StatSensor eGFR versus standard creatinine eGFR.**

| | Standard creatinine estimated GFR | | |
|---|---|---|---|
| StatSensor estimated GFR | Low (< 60) | Normal (≥ 60) | Total |
| Low (< 60) | 52 | 16 | 68 |
| Normal (≥ 60) | 4 | 286 | 290 |
| **Total** | 56 | 302 | 358 |

 

The StatSensor demonstrated high diagnostic performance, with a sensitivity of 92.9% (95% CI: 82.7–98.0), specificity of 94.7% (95% CI: 91.5–96.9), and overall accuracy of 94.4% (95% CI: 91.5–96.6) (Table 3).

The area under the ROC curve (AUC) is 0.938 with standard error of 0.0213, 95% CI (0.896–0.979) p-value <0.001 (Fig 1). The agreement between Stat-Sensor eGFR and Jaffe method based eGFR showed a Kappa value of 0.805 (SE: 0.042, p<0.001) (Table 4).

A scatter plot with linear regression demonstrated a strong positive correlation between the two methods, with a slope of 0.903 and an R² of 0.813. (Fig 2). The Bland–Altman analysis showed that StatSensor-derived eGFR had a **negative bias of 4.36 mL/min/1.73 m²** (95% CI: 3.09–5.64) compared to the reference laboratory Jaffe kinetic creatinine-based

**Table 3. Diagnostic performance of StatSensor for detecting renal dysfunction.**

|  | Estimate | 95% Confidence interval | |
|---|---|---|---|
|  |  | Lower CL | Upper CL |
| Sensitivity (%) | 92.9 | 82.7 | 98.0 |
| Specificity (%) | 94.7 | 91.5 | 96.9 |
| Diagnostic accuracy (%) | 94.4 | 91.5 | 96.6 |
| Positive predictive value (%) | 76.5 | 66.7 | 84.0 |
| Negative predictive value (%) | 98.6 | 96.5 | 99.5 |
| Positive diagnostic likelihood ratio | 17.5 | 10.8 | 28.4 |
| Negative diagnostic likelihood ratio | 0.08 | 0.03 | 0.19 |
| Diagnostic odds ratio | 232.4 | 74.7 | 722.8 |

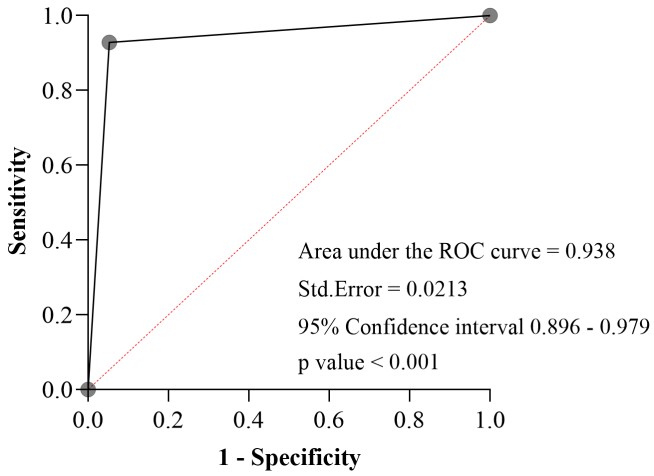

Area under the ROC curve = 0.938

Std.Error = 0.0213

95% Confidence interval 0.896 - 0.979

p value < 0.001

**Fig 1. ROC curve of the StatSensor in detecting renal dysfunction.**

**Table 4. Measure of agreement (Kappa) between StatSensor and creatinine by jaffe method.**

|  |  | Symmetric measures | | | |
|---|---|---|---|---|---|
|  |  | Value | Asymptotic standard error[a] | Approximate T[b] | Approximate significance |
| Measure of Agreement | Kappa | .805 | .042 | 15.342 | < 0.001 |
| N of Valid Cases |  | 358 |  |  |  |

[a]. Not assuming the null hypothesis.

[b]. Using the asymptotic standard error assuming the null hypothesis.

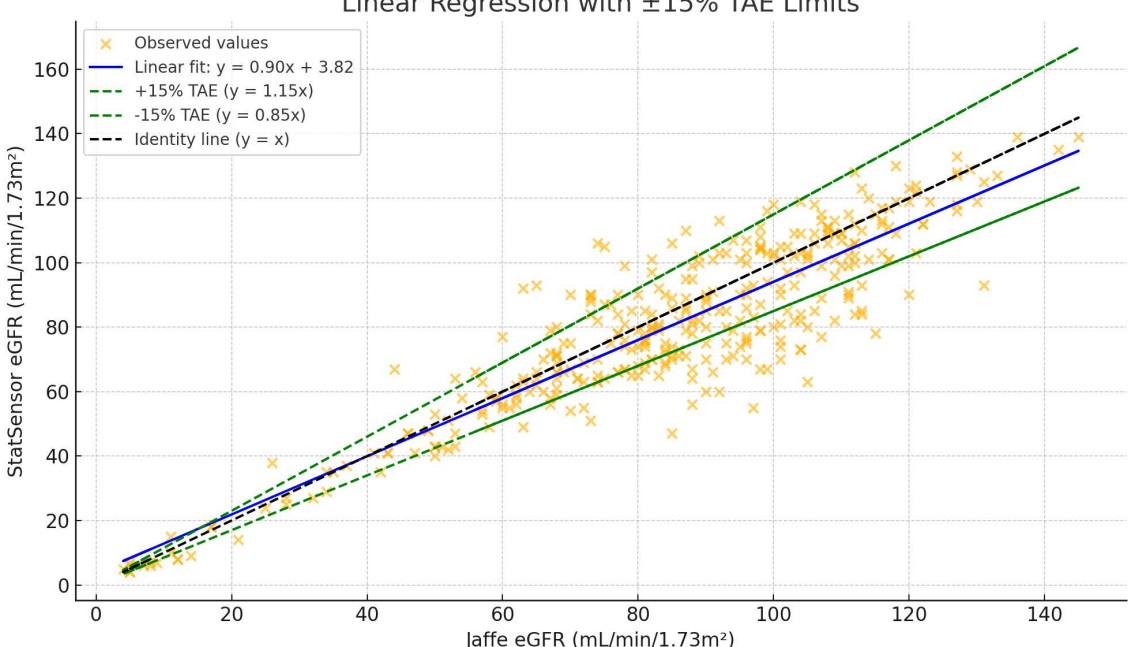

**Fig 2. Linear regression plot of StatSensor eGFR versus Jaffe eGFR with ±15% total allowable error limits.**

eGFR. The 95% limits of agreement ranged from −19.68 to 28.40 mL/min/1.73 m², and the majority of data points fell within the ±15% TEA margin, supporting acceptable agreement between the two methods (Fig 3)

## Discussion

In this cross-section study of individuals receiving HIV care in Tanzania, we demonstrated the accuracy of the StatSensor compared to the conventional serum creatinine method. The StatSensor showed high accuracy in detecting KD, with strong sensitivity, specificity, and diagnostic agreement compared to standard creatinine testing using Jaffe method, with a slight negative bias in eGFR making it effective in distinguishing individuals with and without KD.

This test has the potential to identify 9 out of 10 people with KD correctly. However, 3 out of 10 positive cases may be misclassified, highlighting the need for confirmatory testing for those classified with KD to ensure accurate diagnosis and avoid unnecessary clinical interventions. Our findings align with studies by Chandrasekar et al., and Dally et al., which assessed the accuracy of the StatSensor and reported sensitivity ranging from 85% to 97% and specificity ranging from 90% to 96% and 95%, respectively [11–13]. These results are notably higher than those reported by Nataatmadja et al., likely due to the smaller sample size and fewer events in their study. These values make it an excellent initial screening tool, particularly in primary healthcare settings with limited laboratory infrastructure [15]. By enabling early detection and timely intervention, the StatSensor has the potential to improve clinical outcomes for at-risk patients, especially in resource-limited settings where access to conventional laboratory testing is restricted.

A negative bias of 4.36 mL/min/1.73 m² in eGFR was observed, indicating a under estimation of kidney function when using the StatSensor. This finding is consistent with the study by Kosack et al., which also reported a negative mean eGFR bias when comparing the StatSensor eGFR to laboratory-based eGFR derived from enzymatically measured creatinine [10]. Interestingly, the bias appeared to vary across the eGFR range. At lower eGFR values (<60 mL/min/1.73 m²), there was a mild positive bias, contributing to the misclassification of a few normal cases as abnormal. Conversely,

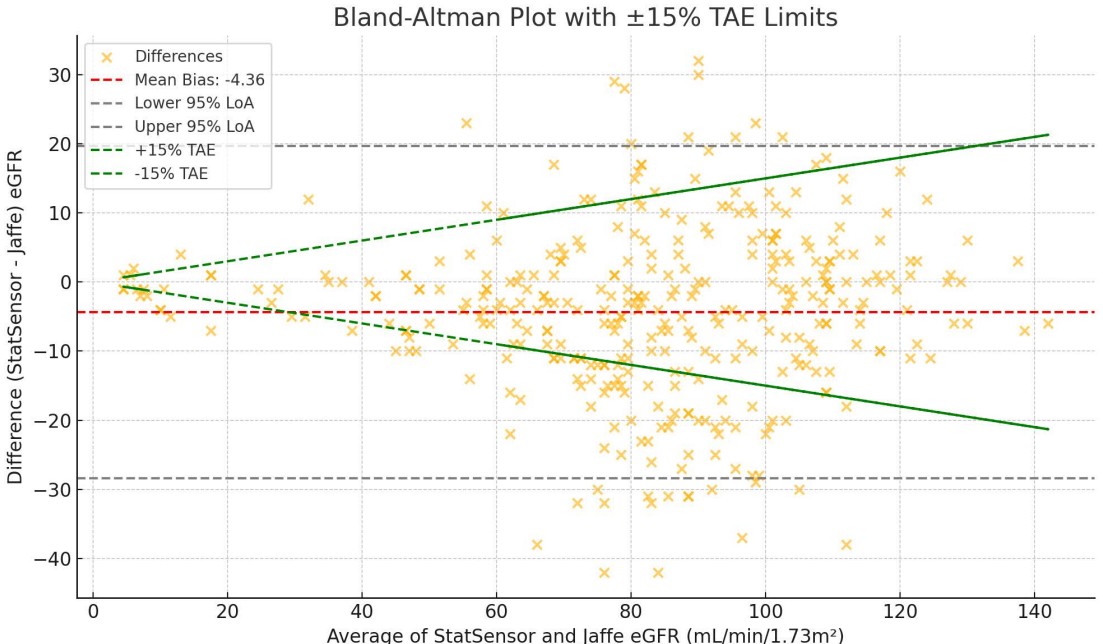

**Fig 3. Bland–Altman plot comparing StatSensor-derived eGFR to laboratory Jaffe method eGFR (mL/min/1.73m²).**

at higher eGFR values, the StatSensor tended to underestimate eGFR, consistent with the overall negative bias, leading to some misclassification of normal as reduced kidney function. These patterns underscore the need to interpret point-of-care results cautiously, especially near diagnostic thresholds. While a mild under-estimation of eGFR is unlikely to cause significant under-diagnosis of KD or delay interventions for patients at risk of CKD progression, it may still be relevant for the misclassification of CKD stages, particularly near diagnostic cutoffs. However, R² alone (0.813) does not fully demonstrate method agreement, as it reflects correlation but not accuracy. To address this, we evaluated the agreement using Bland–Altman analysis and Total Allowable Error (TEa) limits. Over 95% of StatSensor-derived eGFR results fell within the CLSI-recommended ±15% TEa margin, indicating clinically acceptable agreement. Despite the mild bias, the StatSensor demonstrated strong diagnostic performance, similar to previous studies reinforcing its reliability [11,16]. Its ability to provide rapid, point-of-care results makes it particularly valuable for early detection in resource-limited settings where laboratory-based creatinine testing is not always available.

The World Health Organization (WHO) has outlined essential criteria for ideal POC diagnostics in resource-limited settings, summarized by the acronym ASSURED, Affordable, Sensitive, Specific, User-friendly, Rapid and Robust, Equipment-free, and Deliverable to end users [12]. The StatSensor Xpress Creatinine test aligns well with most of these principles, it is portable, easy to operate with minimal training, does not require refrigeration or ancillary laboratory infrastructure, and produces rapid results [11,10].. These diagnostic characteristics, along with its simplicity, make it suitable for use even by non-clinical personnel trained in device handling, thus expanding its applicability in decentralized and primary care settings where access to laboratory services is limited.

Routine and frequent kidney function testing is core part of recommended HIV care. Screening facilitates early detection of KD and enables timely interventions to prevent or slow the progression of disease as well as adjustment of potentially nephrotoxic medication. The StatSensor Xpress Creatinine Meter offers a promising alternative to traditional laboratory-based screening methods, particularly in resource-limited settings where access to conventional testing is limited [9]. Additionally, the device's portability and minimal infrastructure requirements make it particularly suitable for

decentralized healthcare settings, ensuring that KD screening is more accessible to people living with HIV in low-resource environments [9,11].

A major strength of this study is its pragmatic design, conducted within a busy HIV clinic setting in Tanzania, reflecting the real-world feasibility and performance of the StatSensor POC test. The study employed rigorous diagnostic accuracy methods, including blinding and standardized comparisons with the Jaffe method. However, several limitations should be noted. First, the study did not use measured glomerular filtration rate (mGFR), the gold standard for kidney function assessment, which may have introduced classification bias. Second, although the CKD-EPI 2021 equation is widely used, its accuracy in sub-Saharan African populations has been questioned, potentially affecting the precision of eGFR estimates in this cohort. Third, while the StatSensor performed well overall, confirmatory testing may still be warranted near diagnostic thresholds to avoid misclassification. Finally, the study did not evaluate agreement across individual KDIGO eGFR stages, which may have provided additional insights into stage-specific classification discrepancies.

## Conclusion

The StatSensor Xpress Creatinine Meter demonstrated high diagnostic accuracy and a strong correlation with conventional testing, making it a valuable screening tool for KD among PLHIV. Integrating StatSensor into routine HIV care programs could improve patient outcomes by enabling clinicians to adjust ART, implement reno-protective measures, and avoid nephrotoxic drugs. To ensure its effective implementation, further cost-effectiveness analyses are needed to evaluate the feasibility of scaling up its use in national HIV programs.

## Supporting information

**S1 Checklist. A completed STROBE checklist.**
(DOC)

**S2 Data. Dataset of hemodialysis patients.**
(XLSX)

## Author contributions

**Conceptualization:** Daniel Msilanga, Anthony Muiru, Emmanuel Balandya, Rajendra Bhimma, Kathleen Liu.

**Data curation:** Daniel Msilanga, Anthony Muiru, Jonathan Mngumi, Emmanuel Balandya.

**Formal analysis:** Daniel Msilanga, Elizabeth Msangi, Jacqueline Shoo, Paschal Ruggajo, Rajendra Bhimma, Kathleen Liu.

**Funding acquisition:** Daniel Msilanga, Emmanuel Balandya, Kathleen Liu.

**Investigation:** Daniel Msilanga, Anthony Muiru, Emmanuel Balandya, Kathleen Liu.

**Methodology:** Daniel Msilanga, Anthony Muiru, Elizabeth Msangi, Jacqueline Shoo, Jonathan Mngumi, Ewaldo Komba, Emmanuel Balandya, Paschal Ruggajo, Rajendra Bhimma, Kathleen Liu.

**Project administration:** Kathleen Liu.

**Resources:** Kathleen Liu.

**Software:** Anthony Muiru.

**Supervision:** Anthony Muiru, Ewaldo Komba, Kathleen Liu.

**Visualization:** Jacqueline Shoo, Jonathan Mngumi, Kathleen Liu.

**Writing – original draft:** Daniel Msilanga, Anthony Muiru, Elizabeth Msangi.

**Writing – review & editing:** Anthony Muiru, Jonathan Mngumi, Emmanuel Balandya, Paschal Ruggajo, Rajendra Bhimma, Kathleen Liu.

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
