## [Decision Letter · Decision Letter 0]

23 Jun 2025

Dear Dr. Msilanga,

Thank you for submitting your manuscript to PLOS ONE. After careful consideration, we feel that it has merit but does not fully meet PLOS ONE’s publication criteria as it currently stands. Therefore, we invite you to submit a revised version of the manuscript that addresses the points raised during the review process.

We look forward to receiving your revised manuscript.

Kind regards,

Mogamat-Yazied Chothia, MBChB, FCP(SA), MMed, PhD

Academic Editor

PLOS ONE

Journal Requirements:

“This project was funded by the Fogarty International Center of the National Institutes of Health (NIH) Award Number D43TW009343 and the University of California Global Health Institute (UCGHI).”

5. We note you have included a table to which you do not refer in the text of your manuscript. Please ensure that you refer to Table 5 in your text; if accepted, production will need this reference to link the reader to the Table.

Reviewers' comments:

Reviewer's Responses to Questions

**Comments to the Author**

1. Is the manuscript technically sound, and do the data support the conclusions?

Reviewer #1: Partly

Reviewer #2: Yes

2. Has the statistical analysis been performed appropriately and rigorously?

Reviewer #1: No

Reviewer #2: Yes

3. Have the authors made all data underlying the findings in their manuscript fully available?

Reviewer #1: Yes

Reviewer #2: Yes

4. Is the manuscript presented in an intelligible fashion and written in standard English?

Reviewer #1: Yes

Reviewer #2: Yes

Reviewer #1: Thank you to the authors for performing this study. There is a need to confirm performance of point of care tests available. Please address the following concerns.

1. Grammatical error:

a. title should rather read: Point of care creatinine demonstrates high accuracy in detecting kidney dysfunction among people living with HIV in Tanzania

b. Line 58: the sentence does not appear to be complete – a word has been left out after “accessible”?

c. Line 61: “and” should be removed

2. Under the Methods section, the following should be provided:

a. Instrument information required:

i. Methodology in use in the Nova StatSensor Xpress Creatinine Point-of-care meter

ii. Coefficient of variation data at the medical decision limits for the Stat Sensor creatinine measurement

iii. Automated instrument name and model, as well as the correct methodology used for creatinine measurement in the laboratory should be mentioned. The authors mention that the “enzymatic Jaffe method” was used. The method can be either enzymatic or Jaffe (these are two separate methods).

iv. It is also important to mention whether the method is isotope dilution mass spectrometry traceable.

v. Coefficient of variation data at the medical decision limits for the automated creatinine method should also be mentioned.

b. Tube information

i. – “non-heparinized tube” was mentioned. If possible, the authors should be more specific e.g. SST tubes were used

ii. Sample handling information should be summarised e.g. samples were transported to the laboratory and centrifuged at 3000G within 2 hours.

3. Statistical analysis and results

a. As a creatinine analyser is being evaluated, the creatinine data should also be described.

i. In the form of a method comparison using ordinary linear regression and/or difference plots to demonstrate the difference in creatinine results between the methods. Allowable error limits should be included (these can be obtained from EFLM, Westgard or RCPA websites). Remember to include full name of creatinine methods on each axis, as well as correct units used.

b. The eGFR comparison difference plot: the units have not been included along the X or Y-axes. For the X-axis, the laboratory creatinine based eGFR should be used (and not the average eGFR), as this method is being considered the gold-standard method. It is also more useful to demonstrate the allowable error limits on the graphs instead of the 95% distribution limits.

c. eGFR method comparison should ideally be presented before the difference plot graph. Here, as well, please add units on both X and Y axes, as well as the allowable error limits on the graph. The laboratory-based creatinine method should be named on the graph (including the instrument name).

d. Graph legends need to be more descriptive of the findings.

e. Under the results section:

i. Table 1 should omit information not relevant to the study e.g. marital status, level of education and employment status, so that relevant data is easier to read

ii. Lines 123 and 124 are already in table 1 and therefore should be omitted in the paragraph section.

iii. Table 2 data: To be able to get a better idea of the performance of the eGFR results, it would be useful to also compare the KDIGO stages between the two methods in a separate table and then assess % agreement.

iv. Table 3 should be omitted if also being described in lines 135 – 141.

v. In line 153, the positive bias of 4.36 is mentioned. Please add units. This data is more appropriately assessed using total allowable error limits (rather than the limits of agreement alone) – refer to the eGFR difference plot comment.

f. Try to present data in a logical order:

i. Descriptive data

ii. Method comparison and diff plot for creatinine findings

iii. Method comparison and diff plot for eGFR findings

iv. Agreement data (incl kappa), ROC and diagnostic accuracy findings

4. Discussion:

a. Line 173: be clear when discussing the positive bias, whether referring to creatinine or eGFR.

b. Line 186: include the units for eGFR

c. Line 188: This is contradictory to what has been said earlier in the paper. A positive bias for creatinine measurement will mean a negative bias for eGFR measurement.

d. Line 193: Using the R2 value is not enough to confirm good performance between methods. An R2 of 0.813 is not considered a very good correlation. It would be better to demonstrate that the majority of results fall within total allowable error limits (see point above regarding demonstrating the allowable error limits on the graphs)

e. Lines 193 and 218: The authors use the “Jaffe” and “enzymatic” methods interchangeably. This is not correct. Please confirm the exact methodology in use in the laboratory.

This study shows valuable information. Looking forward to seeing the revised version.

Reviewer #2: The Article will contribute to a better understanding of POCT in a resource limited setting

Throughout the article the authors have used the method "enzymatic Jaffe method" for creatinine. There are 2 main methods for creatinine. it is the Jaffe kinetic method or the Enzymatic creatinine method. Please confirm the correct method

Under Methods. Please include the laboratory platform, The CVs of the method used in the laboratory and please add if the lab is engaged in an External Quality assurance programme

Under Results. Please also include the difference plot (Bland Altman) for the 2 creatinine methods. Please ensure that you recognize the hierarchy of methods and label the plot appropriately

The Bland Altman plot for the eGFR needs labelling

Under Discussion: line 173 compared to standard creatinine testing, with a positive bias....

Line 186.... a positive bias of 4.36 eGFR

Please elaborate as to how a positive bias of creatinine is associated with a higher eGFR? Please review

**Do you want your identity to be public for this peer review?** For information about this choice, including consent withdrawal, please see our Privacy Policy

Reviewer #1: No

Reviewer #2: No

---

## [Author Response · Author response to Decision Letter 1]

24 Jun 2025

We confirm that all manuscript and figure files meet PLOS ONE’s formatting and naming requirements.

The PLOS questionnaire on inclusivity in global research has been attached in the supplementary materials submitted with the manuscript.

The statement "The funders had no role in study design, data collection and analysis, decision to publish, or preparation of the manuscript" has been added to the manuscript.

Separate captions for each figure have now been included in the manuscript.

The required information has been incorporated into the manuscript.

The reference list has been thoroughly reviewed and updated to ensure completeness and accuracy.

---

## [Decision Letter · Decision Letter 1]

22 Jul 2025

Dear Dr. Msilanga,

Thank you for submitting your manuscript to PLOS ONE. After careful consideration, we feel that it has merit but does not fully meet PLOS ONE’s publication criteria as it currently stands. Therefore, we invite you to submit a revised version of the manuscript that addresses the points raised during the review process.

We look forward to receiving your revised manuscript.

Kind regards,

Mogamat-Yazied Chothia, MBChB, FCP(SA), MMed, PhD

Academic Editor

PLOS ONE

Journal Requirements:

Reviewers' comments:

Reviewer's Responses to Questions

**Comments to the Author**

Reviewer #1: (No Response)

Reviewer #2: All comments have been addressed

2. Is the manuscript technically sound, and do the data support the conclusions?

Reviewer #1: Yes

Reviewer #2: Yes

3. Has the statistical analysis been performed appropriately and rigorously?

Reviewer #1: Yes

Reviewer #2: Yes

4. Have the authors made all data underlying the findings in their manuscript fully available?

Reviewer #1: Yes

Reviewer #2: Yes

5. Is the manuscript presented in an intelligible fashion and written in standard English?

Reviewer #1: Yes

Reviewer #2: Yes

Reviewer #1: Thank you again for performing this useful study and for addressing the concerns raised. Just a few more concerns, before it can be published:

1. Including the creatinine data would add value to this publication (at least the central data for each method, but even better, would be to also include the linear regression equation (y=0.903x+3.82; r2=0.813) with/without graphs). This can be included in the RESULTS section, just after table 1.

2. Other changes required before publication:

a) Legend for Table 3 (line 183/184) still refers to "laboratory enzymatic eGFR". Should be "Jaffe"

b) Line 154, please add the median (IQR) data for the point of care eGFR results either in Table 1 or in the paragraph describing Table 2 data.

c) eGFR graphs:

* Bland-Altman graph: The TAE error limits have been plotted incorrectly. The TAE is a percentage, whereas

the results are plotted in ml/min/1.73m2. See attached example.

* If possible, please add the TAE limits to the ordinary linear regression graph as well. See attached example.

d) Under the RESULTS and DISCUSSION sections:

* Line 174 – 176:The linear regression relationship demonstration should be described as the equation

(y=0.903x+3.82). In the next line, it is incorrectly stated that a positive bias of 4.36 ml/min/1.73m2 is

present – it appears to be a negative bias (which correlates with the linear regression equation).

* Lines 189: eGFR shows small negative bias.

* Line 202: A negative bias was noted, indicating an underestimation of kidney function, and therefore the

overestimation of low kidney function vs normal function (seen in table 1).

The rest of the discussion and conclusions should be adjusted around this small negative bias and over-calling

of decreased kidney function.

e) Consider changing the title to state "Point-of-care creatinine based eGFR (StatSensor) in detecting kidney dysfunction (KD) among people living with HIV in Tanzania"

f) Adjust abstract according to above changes. In the Results section: An example would be: The StatSensor eGFR correlated with Roche Jaffe method-derived eGFR (y=0.903x+3.82; r2=0.813), with a mean negative bias of 4.36 ml/min/1.73m2.

Reviewer #2: I'm happy with the reviewed changes. Just one note. THE bland Altman plot clearly when truncated shows a positive bias for eGFR at lower ends upto 60 ml/min.This explains the 4 normal missclassfication,however at higher eGfrs there is a negative bias explaining the 16 cases of miss classification. This explanation must be highlighted in the results discussion.

**Do you want your identity to be public for this peer review?** For information about this choice, including consent withdrawal, please see our Privacy Policy

Reviewer #1: No

Reviewer #2: No

---

## [Author Response · Author response to Decision Letter 2]

8 Aug 2025

We sincerely thank the reviewers for their insightful and constructive comments, which significantly helped improve the clarity and rigor of our manuscript.

We have carefully addressed all concerns, including clarifying the bias observed in the Bland–Altman plot, correcting terminologies, and revising the figures and abstract accordingly.

We believe these revisions have strengthened the manuscript and enhanced its value for readers interested in point-of-care kidney function assessment.

---

## [Editor Report · Decision Letter 2]

24 Aug 2025

Point-of-care creatinine-based eGFR (StatSensor) in detecting kidney dysfunction (KD) among people living with HIV in Tanzania

PONE-D-25-28343R2

Dear Dr. Daniel Msilanga,

We’re pleased to inform you that your manuscript has been judged scientifically suitable for publication and will be formally accepted for publication once it meets all outstanding technical requirements.

Kind regards,

Mogamat-Yazied Chothia, MBChB, FCP(SA), MMed, PhD

Academic Editor

PLOS ONE

---

## [Editor Report · Acceptance letter]

PONE-D-25-28343R2

PLOS ONE

Dear Dr. Msilanga,

I'm pleased to inform you that your manuscript has been deemed suitable for publication in PLOS ONE. Congratulations! Your manuscript is now being handed over to our production team.

Kind regards,

on behalf of

Prof. Mogamat-Yazied Chothia

Academic Editor

PLOS ONE